# The Anti-Vitiligo Effects of Feshurin In Vitro from *Ferula samarcandica* and the Mechanism of Action

**DOI:** 10.3390/ph17091252

**Published:** 2024-09-23

**Authors:** Mayire Nueraihemaiti, Zang Deng, Khamidulla Kamoldinov, Niu Chao, Maidina Habasi, Haji Akber Aisa

**Affiliations:** 1State Key Laboratory Basis of Xinjiang Indigenous Medicinal Plants Resource Utilization, Key Laboratory of Chemistry of Plant Resources in Arid Regions, Xinjiang Technical Institute of Physics and Chemistry, Chinese Academy of Sciences, Urumqi 830011, China; mayire0999@sina.com (M.N.); zangdeng@ms.xjb.ac.cn (Z.D.); niuchao@ms.xjb.ac.cn (N.C.); maidn@ms.xjb.ac.cn (M.H.); 2University of Chinese Academy of Sciences, Beijing 100039, China; 3Namangan Engineering-Technological Institute, Namangan 160115, Uzbekistan; xkamoldinov@mail.ru

**Keywords:** vitiligo, melanogenesis, beta-catenin, Janus kinases

## Abstract

Background: Vitiligo is a complex disorder characterized by skin depigmentation; the canonical Wnt signaling pathway that involves *β*-catenin plays a crucial role in promoting the melanin production in melanocytes. Targeted inhibition of the Janus kinase JAK-STAT pathway can effectively diminish the secretion of the chemokine C-X-C motif ligand CXCL10, thereby safeguarding melanocytes. *Ferula* has been applied as a treatment regimen for a long period; however, its use for the treatment of vitiligo has not been previously documented. Methods: CCK-8 assay, Intracellular melanin content assay, Tyrosinase activity assay, Western blotting, qRT-PCR, and ELISA methods were employed. Using molecular docking verified the inhibitory effects of feshurin on the JAK1. Results: The sesquiterpene coumarin feshurin was separated from *Ferula samarcandica*. Feshurin was shown to induce GSK-3*β* phosphorylation, resulting in the translocation of *β*-catenin into the nucleus. This translocation subsequently upregulated the transcription of microphthalmia-associated transcription factor (MITF), leading to increased tyrosinase activity and melanin production. In addition, feshurin inhibited the production of chemokine CXCL10 via the JAK-STAT signaling pathway, which was verified by molecular docking. Conclusions: Based on these findings, it can be concluded that feshurin exhibits significant potential for the development of novel anti-vitiligo therapeutics.

## 1. Introduction

Vitiligo is a complex disorder that manifests as depigmented patches or spots on the skin due to a deficiency of melanin. The etiology of vitiligo is multifactorial, involving various elements, such as genetic predisposition, alterations in the epidermal microenvironment, inherent melanocyte abnormalities, susceptibility to environmental influence, and autoimmune reactions [1]. The occurrence of vitiligo exhibits no significant variation in gender or race and typically manifests between the age of 20 and 30 [2]. Despite its non-life-threatening nature, the conspicuous visibility and persistent nature of vitiligo frequently result in diminished self-esteem, particularly among individuals with darker skin tones, females, and adolescents [3]. Individuals affected by vitiligo experience humiliation and are forced to endure adverse effects on their mental well-being and overall quality of life [4]. Furthermore, the likelihood of vitiligo reoccurrence following treatment cessation has reached 40% [4]. The present approach to the treatment of vitiligo focuses on the prevention of disease onset and the attainment of melanin synthesis [5].

A significant correlation has been observed between melanocytes, tyrosinase (TYR), and the manifestation of vitiligo. In the human skin, melanocytes are situated in the basal layer of the epidermis, with the melanosomes within melanocytes comprising the specific sites required for melanogenesis. The primary role of melanocytes is to produce melanin by oxidizing tyrosine, a process that is carried out by enzymes specific to melanocytes. Melanin is stored in melanosomes and transferred to neighboring keratinocytes to protect the skin from UV radiation. Keratinocytes are essential in maintaining melanocyte homeostasis and melanogenesis, and they form a functional and structural unit [6]. The process of melanin synthesis primarily involves TYR, tyrosinase-related protein 1 (TRP-1), and tyrosinase-related protein 2 (TRP-2) [7]. Melanocyte proliferation, differentiation, and melanogenesis are all aided by these proteins. Among these proteins, TYR acts as a rate-limiting enzyme of melanin synthesis, substantially influencing the rate and quantity of melanin production. Certain transcription factors, specifically microphthalmia-associated transcription factor (MITF), are involved in the melanin synthesis signaling pathway. MITF plays a crucial role by serving as the primary target. Their interaction leads to the upregulation of the TYR expression, which, in turn, facilitates melanin synthesis. Consequently, MITF has emerged as the principal factor responsible for promoting melanin synthesis and serves as the main regulatory factor for melanogenesis [8].

The Wnt signaling pathway plays a crucial role in regulating the expression of MITF. GSK-3*β* is a key enzyme involved in this pathway, and it facilitates the ubiquitination and subsequent degradation of *β*-catenin. Within the Wnt signaling pathway, phosphorylated (p) AKT triggers the phosphorylation of GSK3-*β*, leading to its inhibition. Consequently, *β*-catenin accumulates in the cytoplasm and is subsequently translocated to the nucleus, where it promotes the expression of MITF [9]. Ultimately, the activation of the TYR family of enzymes stimulates melanin synthesis [10,11]. Wenli Lu et al. discovered that inhibiting ZMIZ1 (a coactivator of several transcription factors) expression prevented melanocytes from proliferating, migrating, and invading in vitro. These findings provide compelling evidence that the Wnt signaling pathway is essential in the pathophysiology of vitiligo [12].

Autoimmunity is a significant etiological factor involved in the development of vitiligo. The majority of patients with vitiligo also have other autoimmune diseases such as autoimmune thyroid disease and rheumatoid arthritis [13]. A genome-wide analysis identified the majority of the immunomodulatory genes for vitiligo, including the cluster of differentiation CD44 protein, which plays an important role in T cell development. Abnormally proliferating cytotoxic CD8^+^ T cells have been detected in peripheral blood and lesions have been found in progressive vitiligo. One important source of cytokines and chemokines linked to vitiligo is believed to be keratinocytes [6]. Several cytokines and chemokines play a key role in the development of vitiligo, including IFN-γ. A gene expression analysis of vitiligo lesions has indicated significant upregulation of the expression levels of IFN-γ-induced genes, including the T-cell chemokine receptor CXCR3 and its multi ligands CXCL9 and CXCL10 [14]. Harris et al. demonstrated that IFN-γ-Janus kinase JAK/STAT is an upstream regulatory pathway for CXCL9 and CXCL10. The JAK-STAT pathway is a signal transduction pathway activated by various cytokines and growth factors. This pathway consists primarily of the three following components: tyrosine kinase-related receptors, JAK, and STAT [15]. In vitiligo, IFN-γ is activated by binding to receptors on keratinocytes, which leads to the activation of JAK and the phosphorylation of STAT. P-STAT enters the nucleus causing the transcription of the CXCL9 and 10 genes and further production of the chemokines CXCL9 and CXCL10. Numerous studies have documented that autoreactive CD8^+^ T cells serve as the primary effector cells responsible for the destruction and apoptosis of melanocytes. CD8^+^ T cells in the vasculature are chemotaxed by CXCL9 and CXCL10 and migrate to the epidermis to specifically eliminate melanocytes [16].

The INF-γ-induced JAK-STAT signaling pathway plays a critical role in the expression of CXCL10. Consequently, inhibition of the expression levels of CXCL10 by the JAK-STAT pathway may potentially diminish the recruitment of CD8^+^ T cells and subsequently mitigate the destruction of melanocytes [17]. Ruxolitinib, classified as a small molecule JAK kinase inhibitor, effectively disrupts the JAK-STAT signaling pathway, thereby providing additional suppression of immune-mediated inflammatory pathways [18]. The current approach to treating vitiligo primarily centers on enhancing the melanocyte production of melanin and suppressing autoimmune inflammation such as that induced by the JAK-STAT signaling pathway [19].

*Ferula* has been applied in the treatment of various diseases for a long period of time. Biological and pharmacological studies have confirmed that *Ferula* exhibits antioxidant, antitumor [20], anticoagulant [21], hypotensive [22], hypoglycemic [23,24], analgesic, anti-anxiety, antidepressant [25], analgesic, memory enhancement, neuroprotective [26], anti-spasmodic [27,28], anti-osteoporotic, antibacterial, antimicrobial, and insecticide-like [29] effects. Sesquiterpene coumarins are the main constituents of *Ferula*. It conducts various biological activities, such as those with an anti-inflammatory, antitumor, antioxidant, and antiulcer character [30]. The application of the *Ferula* in vitiligo has not been previously documented. *Ferula samarcandica* is an endemic plant found in Uzbekistan [31]. The compound feshurin, a sesquiterpene coumarin, was separated from the roots of *Ferula samarcandica* and its purity was 97.71%. The present study aimed to assess the melanogenic activity of feshurin and its inhibitory effect on the JAK-STAT signaling pathway in vitro.

## 2. Results

### 2.1. Isolation and Identification of Feshurin

The roots of a *Ferula samarcandica* (2 kg) was extracted at room temperature by EtOH (95%, 10 × 4 L). The ethanolic extract was filtered and concentrated in a rotary evaporator. The dried concentrated extract (218.2 g) was fractionated with various solvents in increasing order of polarity, including hexane, dichloromethane, ethyl acetate, and methanol. The ethyl acetate extract (30.5 g) was subjected to silica gel (200–300 mesh) column chromatography that was eluted with a *n*-hexane/ethyl acetate gradient (8:1–1:1) to yield four fractions (Fr. 1–Fr. 4). Fr. 1 (6.8 g) was subjected to flash column over LiChroprep RP-18 gel, and it was eluted with a gradient of methanol (30–100%) to obtain ten fractions (Fr. 1A–J), and Fr. 1D (503.0 mg) was separated by semi-preparative HPLC to yield pure compound feshurin (98.5 mg).

Feshurin: C_24_H_32_O_5_; colorless crystal; (+)-HRESIMS *m*/*z* 401.2325 [M + H]^+^ (calculated for C_24_H_33_O_5_^+^, 401.2323); ^1^H NMR (600 MHz, CD_3_OD): *δ*_H_ 6.21 (1H, d, *J* = 9.5 Hz, H-3), 7.86 (1H, d, *J* = 9.5 Hz, H-4), 7.50 (1H, d, *J* = 8.6 Hz, H-5), 6.90 (1H, dd, *J* = 8.6, 2.4 Hz, H-6), 6.89 (1H, d, *J* = 2.4 Hz, H-8), 1.56 (1H, m, H-1′*α*), 1.51 (1H, m, H-1′*β*), 1.97 (1H, m, H-2′*α*), 1.51 (1H, m, H-2′*β*), 3.33 (1H, dd, *J* = 11.4, 4.2 Hz, H-3′), 1.41 (1H, d, *J* = 9.5 Hz, H-5′), 1.66 (1H, dd, *J* = 9.6, 3.6 Hz, H-6′*α*), 1.41 (1H, dd, *J* = 12.4, 3.5 Hz, H-6′*β*), 1.78 (1H, dd, *J* = 13.3, 2.9 Hz, H-7′*α*), 1.52 (1H, m, H-7′*β*), 1.49 (1H, dd, *J* = 3.2, 2.4 Hz, H-9′), 4.29 (1H, dd, *J* = 10.2, 2.2 Hz, H-11′*α*), 4.11 (1H, dd, *J* = 10.2, 5.0 Hz, H-11′*β*), 1.17 (3H, s, H_3_-12′), 0.93 (3H, s, H_3_-13′), 0.86 (3H, s, H_3_-14′), and 1.05 (3H, s, H_3_-15′); and ^13^C NMR (150 MHz, CD_3_OD): *δ*_C_ 163.4 (C-2), 113.2 (C-3), 145.8 (C-4), 130.4 (C-5), 114.4 (C-6), 163.9 (C-7), 102.2 (C-8), 157.2 (C-9), 113.9 (C-10), 33.4 (C-1′), 26.0 (C-2′), 76.5 (C-3′), 38.6 (C-4′), 49.7 (C-5′), 19.1 (C-6′), 43.2 (C-7′), 72.9 (C-8′), 59.6 (C-9′), 38.9 (C-10′), 67.6 (C-11′), 31.3 (C-12′), 29.1 (C-13′), 22.8 (C-14′), and 16.7 (C-15′). The absolute structure was the 1′*R*, 2′*R*, 6′*S*, 9′*R*, 10′*S* configuration (Figure 1).

### 2.2. Effects of Feshurin on B16 and HaCaT Cell Viability

The CCK-8 method was employed to assess the cytotoxic effects of feshurin on B16 and HaCaT cells (Figure 2A,B). Following 48 h treatment of B16 cells with feshurin, their cell viability was the following: 94.31 ± 6.20% and 82.98 ± 1.14% at concentrations of 1μM and 10 μM, respectively, and the cytotoxicity of the compounds was increased as their concentration was gradually increased from 10 μM. In the HaCaT cells, cell viability was 125.38 ± 13.76% and 123.4 ± 8.46% at concentrations of 1 μM and 10 μM, respectively. No significant cytotoxicity was observed when the cells were treated with 3 μM of IFN-γ and 10 ng/mL of the JAK kinase inhibitor ruxolitinib alone compared to the concurrent administration of IFN-γ and feshurin. Following investigation with microscopy, no significant effect on cell morphology was noted (Figure 2C,D). Consequently, concentrations ranging from 1 μM to 10 μM were deemed optimal and selected for subsequent experiments.

### 2.3. Intracellular Melanin Content and TYR Activity

The effects of different concentrations of feshurin (1, 5, and 10 μM) on the intracellular TYR activity and melanin production in B16 melanocytes were assessed over 24 and 48 h. The results indicated that (Figure 3A,B) feshurin enhanced melanin synthesis and TYR activity in a concentration-dependent manner. Notably, at a concentration of 5 μM, the melanin content level (174.06 ± 12.03%; *p* < 0.01) was significantly increased compared with that of the control group, while the TYR activity was comparable to that of the positive control group (127.81 ± 10.22%; *p* < 0.01). At a concentration of 10 μM, feshurin significantly increased melanin content to 331.86 ± 9.57% (*p* < 0.01), and the TYR activity was increased to 167.99 ± 7.31% (*p* < 0.01). When B16 cells were treated with feshurin (5 and 10 μM), an increasing number of melanin particles could be observed (Figure 3C). These values were 2.54 times and 1.35 times higher than those noted in the positive control group. Both groups exhibited significantly higher values compared with those of the blank control group, and statistically significant differences were observed.

### 2.4. Feshurin Promotes the Expression Level of Proteins Related to Melanogenesis

To investigate the impact of feshurin on the expression levels of proteins involved in melanin synthesis, Western blot analysis was performed to assess the protein expression levels of TYR, TRP-1, TRP-2, and MITF. The findings revealed a significant enhancement in the expression levels of these proteins following treatment with feshurin (Figure 3D,E). Furthermore, an increase in feshurin concentration corresponded with a proportional increase in protein expression levels. Notably, feshurin (5 or 10 μM) exhibited a significant promotion of all four proteins associated with melanin synthesis. It is important to highlight that the expression levels of TRP-2 reached a significant level at a compound concentration of 1 μM.

### 2.5. Feshurin Mediates Melanogenesis via the Wnt Signaling Pathway

To gain a deeper understanding of the mechanism of melanogenesis mediated by feshurin, Western blot analysis was used to detect the expression of proteins related to the melanogenesis signaling pathway.

Following treatment of B16 cells with feshurin, the levels of p-AKT, GSK-3*β*, p-*β*-catenin-Ser^33,37,41^, and total *β*-catenin exhibited a dose-dependent increase (Figure 4A,B). These findings suggested the involvement of the Wnt signaling pathway in the promotion of melanin production by feshurin. In addition, the expression levels of both cytoplasmic and nuclear *β*-catenin were assessed, and the findings revealed a significantly higher expression of *β*-catenin in the nucleus compared with that of the cytoplasm (Figure 4C,D).

To investigate the impact of the Wnt signaling pathway on the melanogenic effect of feshurin on B16 cells, the GSK-3*β* inhibitor BIO was used to assess melanin production and TYR activity.

Following the treatment of cells with BIO (5 µM), the melanin content and TYR activity were quantified within the cells. The findings of the present study indicated that the concurrent administration of BIO and feshurin resulted in a significant increase in melanin content compared with that noted following treatment of the cells with feshurin alone (10 µM; Figure 4E,F). These results provide additional evidence supporting the notion that treatment of the cells with feshurin leads to the accumulation of *β*-catenin in the nucleus, thereby inducing the expression of MITF and subsequently promoting melanin production.

### 2.6. Feshurin Inhibits the IFN-γ-Induced CXCL10 Production in HaCaT Cells via the JAK-STAT Pathway

Following treatment of the HaCaT cells with various concentrations (1, 3, and 6 μM) of feshurin for 24 h, the mRNA levels of CXCL10 and the chemokine content levels were analyzed using real-time fluorescence quantitative PCR and ELISA, respectively. The findings revealed that the group stimulated with IFN-γ exhibited significantly higher levels of CXCL10 mRNA and other chemokines compared with those of the blank control group. Furthermore, the treatment of cells with feshurin demonstrated a concentration-dependent reduction in CXCL10 mRNA and chemokine levels compared with the IFN-γ stimulation group (Figure 5C,D). These results indicated that feshurin could inhibit the IFN-γ-induced CXCL10 chemokine production in HaCaT cells.

In the present study, Western blot analysis was employed to detect the expression levels of JAK1, JAK2, and the phosphorylation levels of STAT1. HaCaT cells were stimulated with 10 ng/mL IFN-γ, 3 μM of the positive control ruxolitinib (RUX), and varying concentrations (1, 3, and 6 μM) of feshurin for 24 h. This experimental setup allowed the investigation of the effects of feshurin on the JAK-STAT signal transduction pathway. The findings revealed that, in comparison with the blank control group, IFN-γ stimulation significantly activated the phosphorylation of STAT1. However, treatment with feshurin prevented the phosphorylation of STAT1 compared with the IFN-γ stimulation group (Figure 5A,B).

### 2.7. Molecular Docking Studies of Feshurin on JAK1

The docking of feshurin with JAK1 (PDB: 6DBN) produced 10 conformations. By comparatively analyzing the mode of interaction of all the conformations with the protein, the best active conformation for the analysis of non-bonding interactions was selected. Firstly, the active conformation of feshurin was superimposed well with the positive control PF06700841 (also known as Brepocitinib, a JAK1/TYK2 selective inhibitor produced by Pfizer); the benzofuranone ring of the feshurin molecule was placed in the same plane and orientated in the same direction as the pyrimidine A and pyrazole rings of PF06700841 (Figure 6); and a high degree of overlap was noted. The docking score for PF06700841 was −9.23 kJ/mol, and it was −7.36 kJ/mol for feshurin. According to the analysis of the non-bonding interactions, the hydroxy group on the sesquiterpene fragment of feshurin may form hydrogen bonds with amino acids such as Leu881, Asn1008, and Gly882; the corresponding bond lengths were 2.7 Å, 2.8, and 3.0 Å, respectively, and other amino acids mainly interacted with the ligand through hydrophobic interactions. In addition, the hydrogen–bond interaction between benzopyrone (coumarin fragment) and residues surrounding the active site was not found, which explained the lower score of feshurin compared with that of PF06700841.

## 3. Discussion

The etiology of vitiligo is intricate and multifactorial. Numerous hypotheses, primarily focusing on autoimmunity, oxidative stress, melanocyte self-destruction, and other related factors, have been proposed by researchers regarding its etiology. Given the intricate nature of its mechanism, there is currently no pharmacological intervention that is both safe and efficacious in curing vitiligo [32]. Consequently, the exploration and investigation of lead compounds that can be used for the treatment of vitiligo are of significant importance.

Melanin synthesis is primarily regulated by proteins from the TYR family and the transcription factor MITF [33]. In the present study, the sesquiterpene coumarin feshurin was isolated from *Ferula samarcandica*. The findings demonstrated that feshurin exhibited a significant melanogenesis-promoting effect and induced the expression levels of TYR family proteins (TYR, TRP-1, TRP-2, and MITF) in a concentration-dependent manner.

The Wnt pathway, activated by the Wnt glycoprotein family, plays a crucial role in intercellular signaling. *β*-catenin is the key protein involved in this pathway [34]. Activation of the Wnt pathway leads to a negative regulation of GSK-3*β*, resulting in the formation of a complex between *β*-catenin, lymphocyte enhancer factor 1, and T cell factor. This complex leads to the accumulation of *β*-catenin and its subsequent translocation into the nucleus, ultimately leading to the upregulation of MITF expression [10].

Following the treatment of B16 cells with feshurin, a concentration-dependent increase was noted in the expression levels of phosphorylated AKT, GSK-3*β*, *β*-catenin-Ser^33,37,41^, and total *β*-catenin. These findings suggest the involvement of the Wnt signaling pathway in the promotion of melanin production by feshurin. In addition, the expression levels of both cytoplasmic and nuclear *β*-catenin were assessed during melanin synthesis. The results revealed an appreciably higher expression of *β*-catenin in the nucleus compared with that noted in the cytoplasm.

The JAK-STAT signaling pathway has been implicated in the pathogenesis of autoimmune and inflammatory diseases. In addition, it plays a crucial role in various significant biological processes, including cell differentiation, proliferation, apoptosis, and immune regulation. Numerous cytokines are known to participate in this pathway [35,36].

The chemokine CXCL10, induced by IFN-γ, is produced by various cell types, including neutrophils, monocytes, and keratinocytes [37]. In the pathogenesis of vitiligo, CXCL10 plays a crucial role in the initiation and maintenance of skin depigmentation [38,39]. Keratinocytes, specifically HaCaT cells, are considered the primary source of CXCL10 secretion [40]. Following IFN-γ binding to its receptor on keratinocytes, JAK is activated, leading to downstream STAT phosphorylation. The phosphorylated STAT is translocated into the nucleus, promoting the transcription of the CXCL10 gene and the subsequent production of the chemokine CXCL10. Furthermore, the chemotaxis of CD8^+^ T cells in the vasculature, induced by CXCL10, plays a significant role in the migration of keratinocytes to the epidermis for melanocyte destruction [15]. This phenomenon has gained attention in the medical field, leading to the recent approval of Opzelura™ (RUX) by the FDA (U.S. Food and Drug Administration) for the treatment of non-segmental vitiligo [41,42].

In the present study, it was discovered that feshurin facilitates GSK-3*β* phosphorylation, resulting in the translocation of *β*-catenin into the nucleus. This translocation subsequently upregulates MITF transcription, ultimately leading to an increase in TYR activity and melanin production. This conclusion was further substantiated through specific inhibitor experiments targeting the signaling pathway. It was also observed that feshurin could inhibit IFN-γ-induced CXCL10 expression in HaCaT cells, while also preventing the phosphorylation of STAT1. The gradual increase in feshurin concentration was associated with a corresponding decrease in the phosphorylation levels of STAT1. The findings indicated that feshurin could diminish the expression levels of CXCL10 induced by IFN-γ via the inhibition of the JAK-STAT signaling pathway. Molecular simulation demonstrated that feshurin may inhibit JAK1 via hydrogen bonding and hydrophobic interactions. Based on the experimental results, feshurin exhibited promising potential for the development of novel anti-vitiligo medications.

## 4. Materials and Methods

### 4.1. Plant Material

The roots of *Ferula samarcandica* were collected on September 2022 from the Surkhandarya region, Uzbekistan, and they were identified by Prof. Alim Magnurovich Nigmatullayev, Institute of the Chemistry of Plant Substances, Academy of Sciences of the Republic of Uzbekistan. The herb was deposited as a voucher specimen (ICPS2229).

### 4.2. Reagents

The following reagents were used: methanol, n-hexane, ethyl acetate, and ethanol (analytical purity, Tianjin Kemiou Reagent Co., Jin Nan Qu, China); acetonitrile (chromatographic purity, Merck Reagent, Rahway, NJ, USA); DMEM high-glucose medium (Gibco; Thermo Fisher Scientific, Inc.; REF:C11995500BT, Lot: 8120490, Waltham, MA, USA); dimethyl sulfoxide (DMSO, Sigma-Aldrich; Merck KGaA; V900090-500 mL; St. Louis, MO, USA); 0.25% trypsin-EDTA (Gibco; Thermo Fisher Scientific, Inc., REF:25200-056, LOT:2466757, Waltham, MA, USA); penicillin streptomycin solution (HyClone; Cytiva, Lot:J220021, Logan, UT, USA); Trypanblue (Invitrogen; Thermo Fisher Scientific, Inc.; REF:T10282, Lot:2117648, Waltham, MA, USA); FBS (VivaCell, REF:C04001-500, LOT:2238253, Yerevan, Armenia); 8-methoxypsoralen (Sigma-Aldrich; Merck KGaA, Lot: MKCC4366, St. Louis, MO, USA); 4-nitrophenyl phosphatedisodium salt hexahydrate (MACKLIN, CAS:333338-18-4, LOT:C12082780, Shanghai, China); sodium fluoride (Sigma-Aldrich; Merck KGaA, CAS:7681-49-4, St. Louis, MO, USA); phenylmethanesulfonyl fluoride (Sigma-Aldrich; Merck KGaA, CAS:329-98-6, St. Louis, MO, USA); OV (CAS:13721-39-6, Beijing Solarbio Science & Technology Co., Ltd., Beijing, China); DL-dithiothreitol (Sigma-Aldrich; Merck KGaA; LOT:SLBB5467V, St. Louis, MO, USA); benzamidine (Sigma-Aldrich; Merck KGaA, CAS:618-39-3,LOT:BCBL718V, St. Louis, MO, USA); 3,4-dihydroxy-L-phenylalanine (L-DOPA; Sigma-Aldrich; Merck KGaA, CAS:59-92-7, Lot:SLCB0627, St. Louis, MO, USA); 6-bromoindirubin-3′-oxime (BIO; AMQUAR Biology, Shanghai, China); bicinchoninic acid (BCA) protein assay kit (Thermo Fisher Scientific, Inc.; REF:23227,LOT:WL344130, Waltham, MA, USA); TritonX-100 (Sigma-Aldrich; Merck KGaA, CAS:9002-93-1,LOT:WXBB7764V, St. Louis, MO, USA); TRNzol Universal RNA Reagent (Tiangen Biotech Co., Ltd., LOT:Y1321, Beijing, China) and SYBR Green PCR kit (Qiagen AB; cat no. 208054, Düsseldorf, Germany); the antibodies against TYR (SC7833), TRP-1 (SC25543), and TRP-2 (SC25544) were obtained from Santa Cruz Biotechnology, Inc., Dallas, TX, USA; the antibodies against p-GSK-3*β* (cat. no. #5558S), GSK-3*β* (cat. no. #9832S), p-*β*-catenin (Ser33,37,41 cat. no. #4511s), *β*-catenin (cat. no. #8480S), p-AKT (cat. no. #13038T), AKT (cat. no. #9272), MITF (cat. no. #97800s), p-P38 (cat. no. #4511S), P38 (cat. no. #9212S), p-CREB (cat. no. #9196S), CREB (cat. no. #9104S), p-ERK (cat. no. #9106S), ERK (cat. no. #4696s), JAK-1 (cat. no. #3344S), JAK-2 (cat. no. #3230S), p-STAT1 (cat. no. #9167S), p-JNK (cat. no. #4668S), and JNK (cat. no. #9252S) were obtained from Cell Signaling Technology, Inc. (Shanghai, China); and the antibodies against lamin B (JB10) and the suppressor of cytokine signaling 1 (cat. no. A23E03) were from ABS.

### 4.3. Cell Culture

B16 melanoma cells were purchased from the Beijing Chinese Academy of Sciences (RRID: CVCL-F936). HaCaT cells (keratinocytes) were obtained from the Wuhan China Center for Type Culture Collection (RRID: CVCL-0038). The cells were cultured with DMEM containing 10% fetal bovine serum, 1% penicillin (100 U/mL), and streptomycin (100 mg/mL) in a constant temperature incubator with 5% CO_2_ at 37 °C.

### 4.4. Cell Viability Assay

Cell viability was assessed using the Cell Counting Kit-8 (CCK8) assay. The cells were placed at a density of 5 × 10^3^ cell/well, with a volume of 100 µL, seeded into 96 well plates, and incubated in a constant temperature incubator containing 5% CO_2_ at 37 °C for 24 h; the medium was discarded and 100 µL of fresh culture medium containing different concentrations of feshurin to be tested was added to the cells for an additional 48 h. The same volume of untreated cells as the blank control was used. Following 48 h, 10 µL of CCK8 solution was added to each well and the cells were incubated for 2 h. The absorbance value was measured at 450 nm [Cell viability = feshurin treatment group optical density (OD)_450_/blank control group OD_450_ × 100%].

### 4.5. Determination of Relative Intracellular Melanin Content

B16 melanoma cells seeded in six-well plates (2 × 10^5^ cells) were incubated in a constant temperature incubator in the presence of 5% CO_2_ at 37 °C; the cells were allowed to attach to the well, the medium was discarded, and the culture medium was added containing different concentrations of feshurin for an additional 48 h. Subsequently, the cells were lysed with RIPA lysate and transferred to a 1.5 mL Eppendorf tube. The cells and lysate were thoroughly mixed by rotation for 40 min. The samples were centrifuged at 12,000 rpm for 22 min, and the protein content of the cell supernatant was measured using the BCA method. A total of 200 µL of NAOH/5% DMSO solution was added to the cell precipitate, which was dissolved at 80 °C for 1 h, and the absorbance value was measured at 405 nm [(relative melanin content = Treatment group OD_405_/blank control OD_405_) × 100%].

### 4.6. Determination of Intracellular TYR Activity

The intracellular TYR activity was measured using the L-DOPA method. B16 melanoma cells were seeded in six-well plates (3.5 × 10^5^) and incubated at a constant temperature incubator in the presence of 5% CO_2_ at 37 °C. Following attachment of the cells to the well, they were further cultured in a medium containing different concentrations of feshurin for 24 h. Cell lysis was performed using a solution containing 1% sodium deoxycholate + 1% Triton X-100, and it was then placed at −80 °C for 30 min and centrifuged at 12,000 rpm for 22 min. The protein content of the cell supernatant was measured using the BCA method. The supernatant samples (50–70 µL) were incubated with 10 µL of L-DOPA at 37 °C for 20 min and the absorbance value was measured at 490 nm. Three wells were set for each concentration and the experiment was repeated three times to estimate the average value.

### 4.7. Western Blotting

The cell lysate was collected according to the detection method of the intracellular melanin content. The supernatant was obtained, and the protein concentration was measured; the protein sample was denatured in the buffer, separated with 10% SDS-PAGE, and transferred to a polyvinylidene fluoride membrane. The membrane was incubated with a 5% skim milk powder solution for 1 h and further incubated overnight with a primary antibody used at a dilution of 1:1000 at 4 °C. Following washing with TBST (Tris Buffer Saline containing 1% Tween 20), the membrane was incubated with a secondary antibody (1:2000 dilution) at room temperature for 1 h. The membrane was washed with TBST three times, and ECL protein blotting detection reagent (GE Healthcare; Cytiva, USA) was used to reveal the protein bands. The nuclear/cytoplasmic protein separation experiment was conducted according to the instructions of the nuclear/cytoplasmic separation kit, and the proteins were collected separately for use. The Adobe Photoshop (Adobe Systems, Inc., USA) program was used for quantitative analysis of the protein bands.

### 4.8. Real-Time Reverse Transcription-Quantitative PCR (RT-qPCR) (TaqMan) ™ Analysis

TRIzol^®^ Reagent (Invitrogen; Thermo Fisher Scientific, Inc.) was used to isolate the total RNA from the feshurin-treated HaCaT cells. A spectrophotometer was used to determine the purity and concentration of the total RNA; the optimal absorbance (260/280 nm) of the RNA sample ranged between 1.8 and 2.0. According to the instructions provided by the manufacturer, the Prime Script RT reagent Kit (Tiangen Biotech Co., Ltd.) was used to convert the total RNA into cDNA for RT-qPCR. The primer design and synthesis were performed by referring to studies in the literature, and the primer sequences were synthesized by ShengGong Biotechnology Co., Ltd., China (Table 1). Fluorescence quantitative detection was performed on an Applied Biosystems 7300 PCR system (Applied Biosystems; Thermo Fisher Scientific, Inc.) thermal cycling instrument. All experiments were performed in three replicates. According to the threshold (Ct) of the experimental and control groups, the calculation of the relative mRNA expression levels of each experimental group was performed using the 2^−ΔΔCq^ method.

### 4.9. Enzyme-Linked Immunosorbent Assay (ELISA)

The CXCL10 kit was purchased from Absin, China. Different concentrations of compounds were used to stimulate the keratinocytes (HaCaT) for 24 h, and the detection of the CXCL10 chemokine levels in the cell supernatant was strictly performed according to the instructions of the ELISA kit.

### 4.10. Molecular Docking

Molecular docking studies were performed using Discovery Studio 2019. The PDB format files of the JAK1 proteins were retrieved from RCSB (https://www.rcsb.org/). Water molecules and inorganic ions were removed. The proteins were prepared for processing (Tools-Macromolecules-Prepare Protein-Clean Protein; Tools-Macromolecules-Prepare Protein-Prepare Protein-Run). The binding sites were identified in the receptor cavities (Tools-Receptor Ligand Interactions-Define and Edit Binding Site-From Receptor Cavities). The processed proteins were saved as SDF files. The ligand molecules were also saved as SDF files. Structural optimization of the ligand molecules (Tools-Small Molecules-Minimize Ligands-Full Minimization) was performed. The CDOCKER module of DS was used for molecular docking (Tools-Receptor Ligand Interactions-Dock ligands-CDOCKER), visualization, and the analysis of the results.

### 4.11. Statistical Analysis

All experiments were carried out in triplicate, and all the data conforming to normal distribution were expressed as the mean ± standard deviation. GraphPad Prism 9.0.0 software (GraphPad Software, Inc., San Diego, CA, USA) (RRID: SCR-002798) was used to conduct a statistical analysis of the results. One-way ANOVA and Tukey’s multiple comparison tests were used to compare the statistical differences between the treatment groups. A *p* ≥ 0.05 was considered not significantly different, a * *p* < 0.05 was considered statistically different, a ** *p* ≤ 0.01 was considered statistically significantly different, and a *** *p* ≤ 0.001 and a **** *p* ≤ 0.0001 were considered extremely significantly different.

## Figures and Tables

**Figure 1 pharmaceuticals-17-01252-f001:**
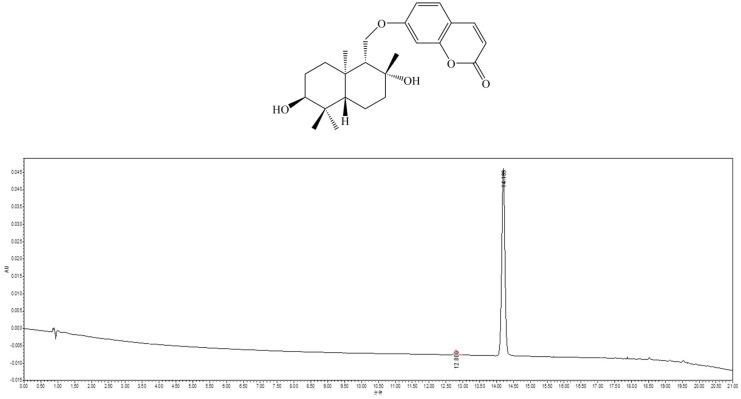
The chemical structure and ultra−performance liquid chromatogram of feshurin (detection wavelength at 254 nm). The Chinese for the horizontal axis is minute.

**Figure 2 pharmaceuticals-17-01252-f002:**
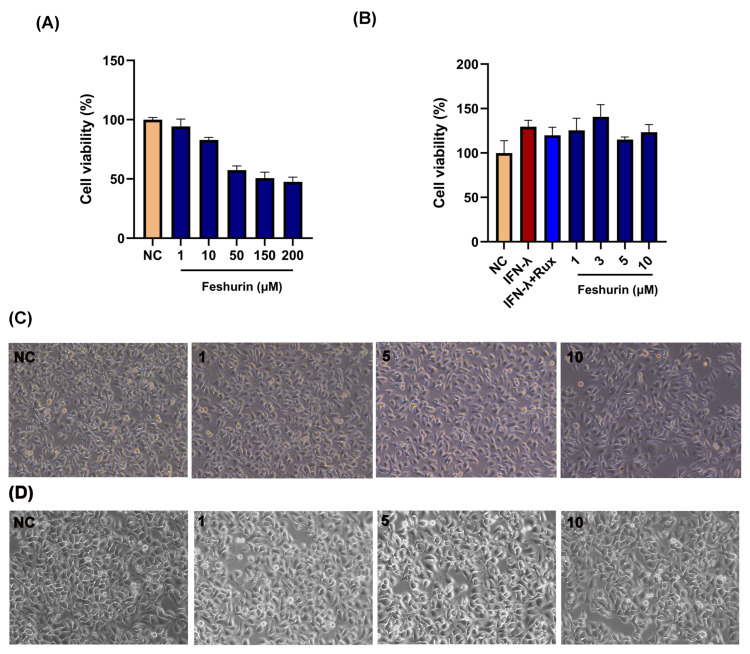
Effects of feshurin on cell viability and morphology. (**A**,**B**) The effect of different concentrations (0–200 µM) of feshurin on B16 melanoma and HaCaT cells for 48 h. Cell viability was measured by the CCK-8 assay. (**C**,**D**) The morphological changes in the feshurin-treated (0–10 µM) B16 cells and HaCaT cells, as determined by microscopy (100× magnification). (C-NC and D-NC) 0.1% DMSO, (C-1 and D-1) 1 µM, (C-5 and D-5) 5 µM, and (C-10 and D-10) 10 µM of feshurin. CCK-8, cell counting kit-8; DMSO, dimethylsulfoxide; NC, negative control; IFN-γ, interferon-γ; and Rux, ruxolitinib.

**Figure 3 pharmaceuticals-17-01252-f003:**
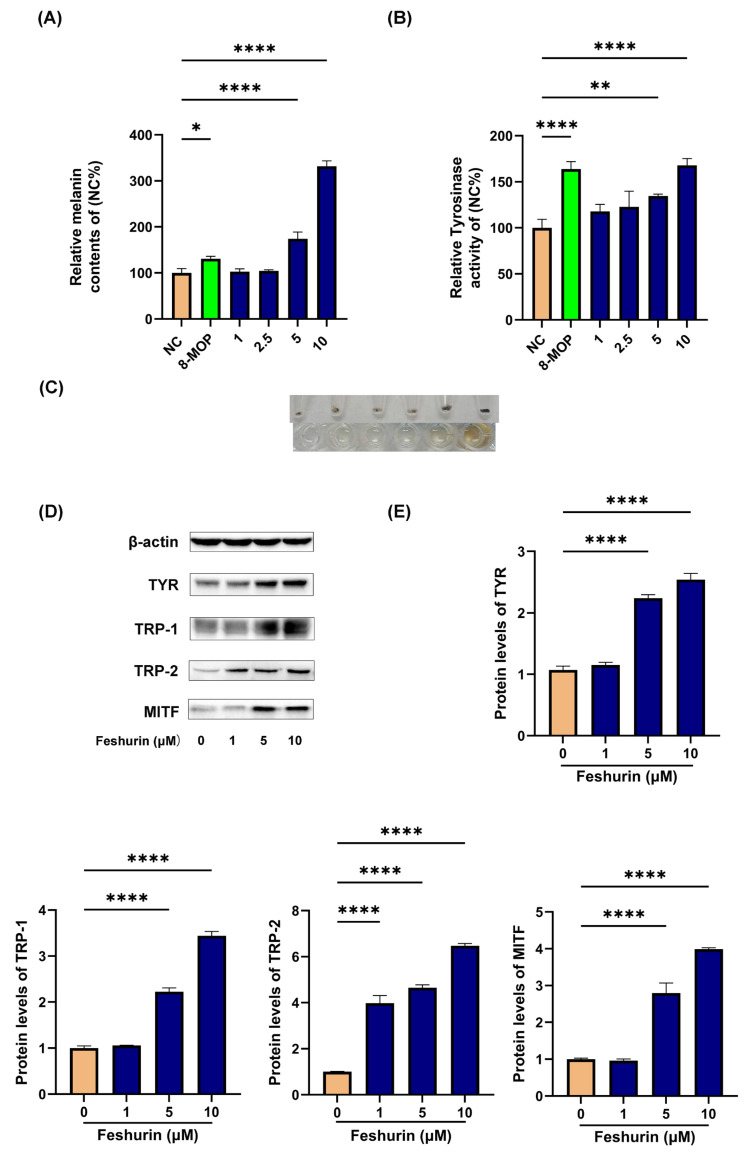
Effects of feshurin on TYR activity, melanin content, and the melanogenesis-related genes. (**A**) Presentation of the melanin content following 48 h of treatment with (0–10 μM) of feshurin. (**B**) TYR activity following 24 h feshurin treatment. (**C**) Representation of the image indicating the cell precipitation and dissolved solution in the presence of 5% sodium hydroxide. (**D**,**E**) The protein levels of MITF, TYR, TRP-1, and TRP-2 were detected following 48 h of feshurin treatment. Photoshop (Adobe Systems, Inc.) was used to determine the protein band density (* *p* < 0.05, ** *p* < 0.01, **** *p* < 0.0001). The data are shown as the mean ± SD and were analyzed by one-way ANOVA followed by Tukey’s test. All experiments were performed three times. TYR, tyrosinase; MITF, microphthalmia-associated transcription factor; and TRP, tyrosinase-related protein.

**Figure 4 pharmaceuticals-17-01252-f004:**
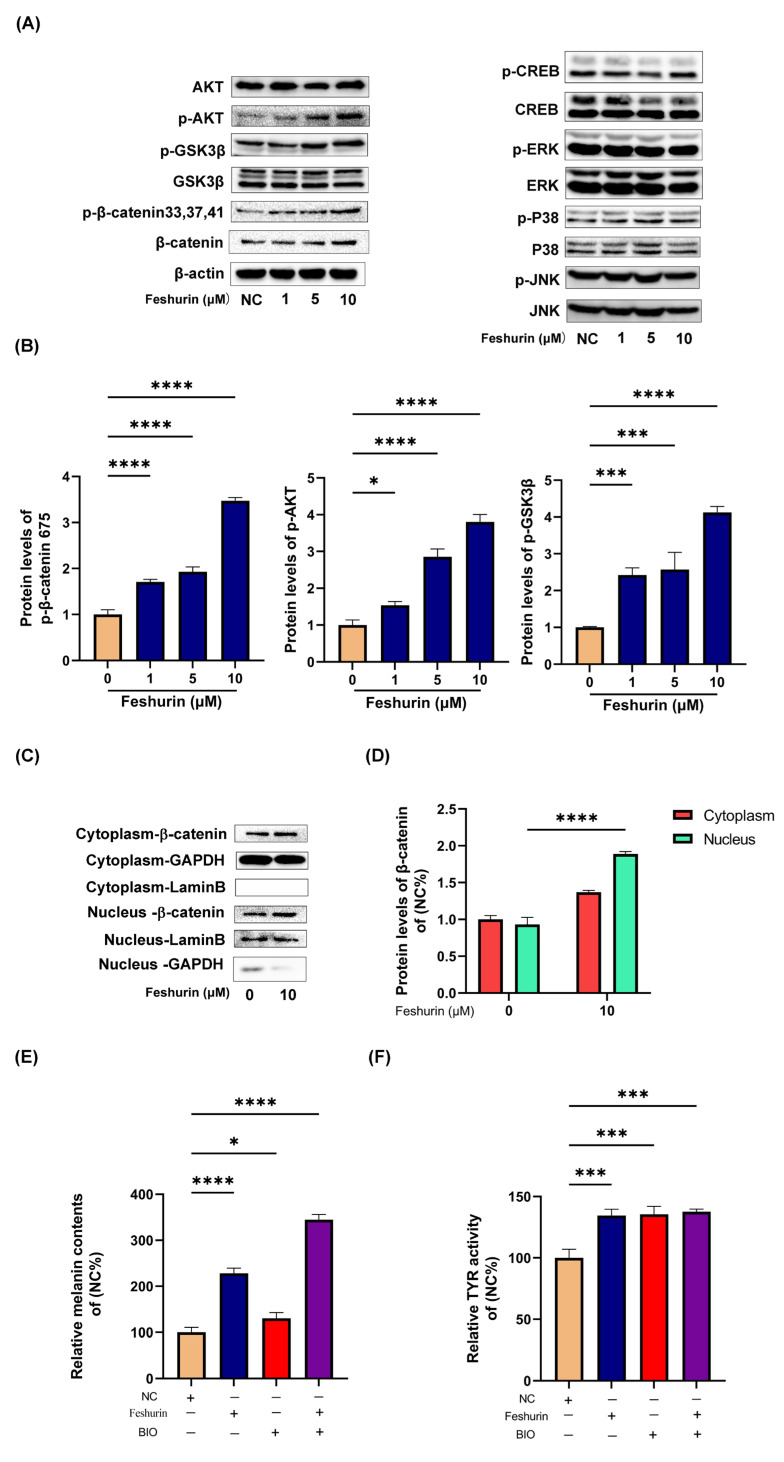
The compound feshurin activated the Wnt signaling pathway via the phosphorylation of GSK-*3β*, *β*-catenin, and AKT. (**A**,**B**) The expression levels of melanogenesis-related signaling pathway proteins in the B16 cells treated with (0–10 μM) feshurin for 48 h were analyzed compared with those of *β*-actin. (**C**,**D**) Following the treatment of the B16 cells with DMSO or feshurin (10 μM) for 24 h, the nucleus and cytoplasmic *β*-catenin expression levels were observed. The cytoplasmic protein levels were standardized against the expression levels of GAPDH and those of the nuclear proteins against the expression levels of lamin B. (**E**,**F**) B16 cells were pre-treated with BIO (5 µM) for 2 h and subsequently incubated with feshurin (10 µM) for 24 h to measure the melanin content and TYR activity. Photoshop (Adobe Systems, Inc.) was used to determine the protein band density (* *p* < 0.05, *** *p* < 0.001, **** *p* < 0.0001). The data are shown as the mean ± SD and were analyzed by one-way ANOVA followed by Tukey’s test. All experiments were performed three times. DMSO, dimethylsulfoxide; TYR, tyrosinase; BIO, 6-bromoindirubin-3′-oxime; and SD, standard deviation.

**Figure 5 pharmaceuticals-17-01252-f005:**
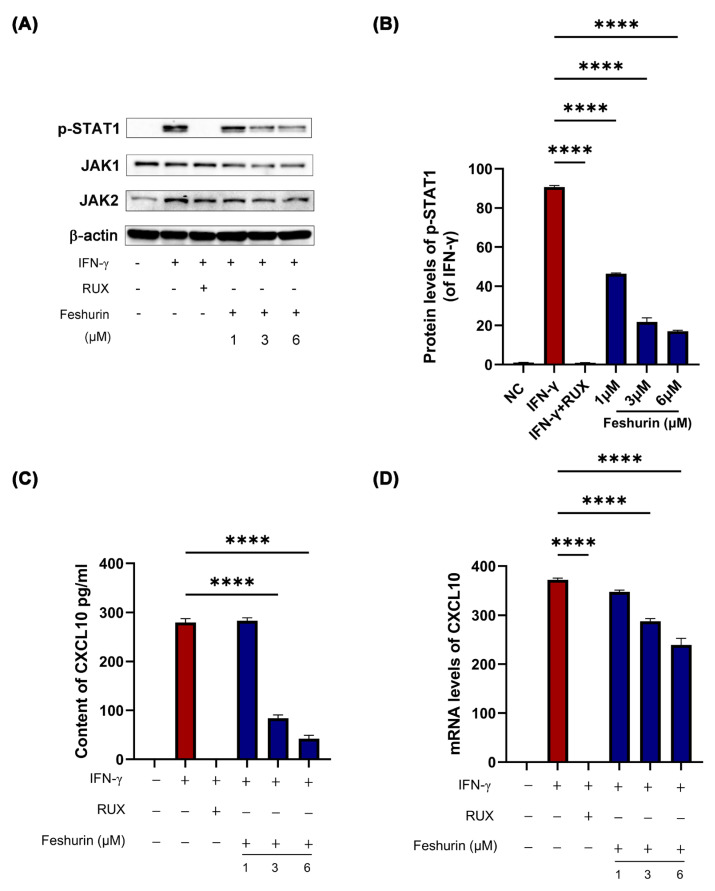
Feshurin inhibits the IFN-γ−induced CXCL10 production in HaCaT cells via the JAK-STAT pathway. (**A**) Following treatment of the HaCaT cells with different concentrations (1–6 μM) of feshurin for 24 h, the expression levels of p-STAT1, JAK-1, and JAK-2 were observed. (Ruxolitinib was used as a positive control and STAT1 phosphorylation was stimulated by IFN-γ.) (**B**) The expression levels of p-STAT1 were analyzed and compared with those of *β*-actin. Photoshop (Adobe Systems, Inc.) was used to determine the protein band density. (**C**) The content of CXCL10 in the cell supernatant was detected by ELISA. (**D**) RT-qPCR was used to detect the mRNA expression levels of CXCL10. Photoshop was used to determine the protein band density (**** *p* < 0.0001). The data are shown as the mean ± SD and were analyzed by one-way ANOVA followed by Tukey’s test. All experiments were performed three times. CXCL, chemokine (C-X-C motif) ligand; p-STAT1, phosphorylated-STAT1; JAK, Janus kinase; ELISA, enzyme-linked immunosorbent assay; RT-qPCR, reverse transcription-quantitative PCR; and SD, standard deviation.

**Figure 6 pharmaceuticals-17-01252-f006:**
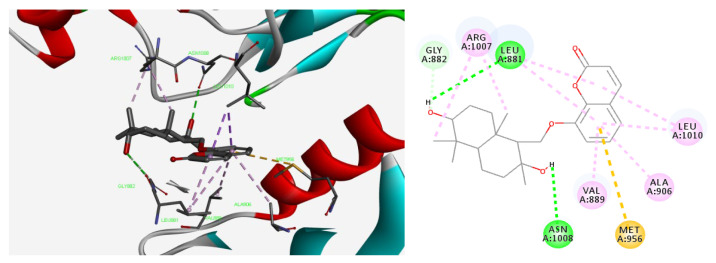
Molecular docking studies were performed to examine the effects of feshurin on JAK1. A 2D and 3D diagram of non-bonding interactions between feshurin and JAK1. JAK, Janus kinase.

**Table 1 pharmaceuticals-17-01252-t001:** The primer sequence of the target genes for real-time PCR.

Primer	Forward(5′-3′)	Reverse(5′-3′)
*CXCL-10*	GTGGCATTCAAGGAGTACCTC	TGATGGCCTTCGATTCTGGATT

## Data Availability

The data that support the findings of this study are available from the corresponding authors upon reasonable request.

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
