# Peer review of "The Anti-Vitiligo Effects of Feshurin In Vitro from Ferula samarcandica and the Mechanism of Action"

_pharmaceuticals, 2024, doi:10.3390/ph17091252_

Round 1

Reviewer 1 Report

Comments and Suggestions for Authors

The manuscript presents a novel investigation into the anti-vitiligo effects of feshurin, a compound derived from Ferula samarcandica. The study aims to elucidate the mechanism of action, particularly focusing on the Wnt signaling pathway and the JAK-STAT pathway's role in melanogenesis. The research is timely and relevant, given the increasing interest in natural compounds for treating skin disorders.

Recommendations for Authors

The manuscript would benefit from more structured sections, particularly in the Results and Discussion areas. Clear headings and subheadings can enhance readability.

While the study reports significant findings, the statistical analyses should be better detailed. Include specifics on the statistical methods used and the rationale for chosen significance thresholds.

The introduction could be strengthened by incorporating more recent studies to provide a broader context for the research.

Some figures lack clear legends and explanations. Ensure all figures are adequately described to aid in the reader's understanding.

Ensure consistent formatting throughout the manuscript.

Check for grammatical errors and typos, especially in the abstract and introduction.

Clarify the concentrations used in the experiments and their significance.

I recommend Major Revisions. The authors should address the issues related to clarity, statistical analysis, and literature context. A revised manuscript that incorporates these recommendations would significantly enhance the quality and impact of the study.

Comments on the Quality of English Language

Check for grammatical errors and typos, especially in the abstract and introduction.

Author Response

Comments 1: The manuscript would benefit from more structured sections, particularly in the Results and Discussion areas. Clear headings and subheadings can enhance readability.

Response 1: We would like to thank you for your careful reading, helpful comments, and constructive suggestions, we've added subheadings to the results section of the article. Which have significantly improved the presentation of our manuscript.

Comments 2: While the study reports significant findings, the statistical analyses should be better detailed. Include specifics on the statistical methods used and the rationale for chosen significance thresholds.

Response 2: We thank the reviewer for this valuable suggestion. We have added the statistical methods used and the rationale for chosen significance thresholds. (Page 16, line 458-465) marked in red.

All experiments were carried out in triplicate, and all data conforming to normal distribution are expressed as mean ± standard deviation. GraphPad Prism 9.0.0 software (GraphPad Software, Inc.) (RRID: SCR-002798) was used to conduct statistical analysis of the results. One-way ANOVA and Tukey’s multiple comparison tests were used to compare the statistical differences between the treatment groups. P ≥ 0.05 was considered not significantly different, P*<0.05 was considered statistically different, **p ≤ 0.01 was considered statistically significantly different, ***p ≤ 0.001, ****p ≤ 0.0001 was considered extremely significantly different.

Comments 3: The introduction could be strengthened by incorporating more recent studies to provide a broader context for the research.

Response 3: As the Reviewer's good advice, we added more recent studies in the manuscript's introduction according to your thoughtful comments.

(Page 1, line 44-45, Page 2, line 46-51,65-68,75-76,94-96) marked in red.

Comments 4: Some figures lack clear legends and explanations. Ensure all figures are adequately described to aid in the reader's understanding.

Response 4: Thank you for your valuable advice. We have added legends and descriptions to some of the figures: Figure 1 (Page 4, line 138-139), Figure 2 (Page 5, line 155-160); Figure 3 (Page 7, line 187, Page 8, line 191); Figure 4 (Page 10, line 220-221); Figure 5 (Page 11, line 252)

Comments 5: Clarify the concentrations used in the experiments and their significance.

Response 5: Thank you for your valuable advice. At a concentration of 5μM, the melanin content level was significantly increased compared to the control group (174.06 ± 12.03%; P<0.01), the tyrosinase activity was close to the positive control group (127.81 ± 10.22%; P<0.01). At a concentration of 10μM, the melanin content significantly increased to 331.86 ± 9.57% (P<0.01), and the tyrosinase activity increased to 167.99 ± 7.31% (P<0.01). These values were 2.54 times and 1.35 times higher than the positive control group. To highlight the compound's activity and the concentration-dependent relationship between the compounds' activity, we used concentrations of 1µM, 5µM, and 10 µM for further study in B16 cells, 1µM, 3µM, and 6 µM in HaCaT cells.

Reviewer 2 Report

Comments and Suggestions for Authors

1.      Page 4, figure 2B: Authors should explain or comment regarding increase of activity in CCK-8 assay after Addition of feshurin in  HaCaT cell. How this experiment contribute to Anti-Vitiligo Effects of Feshurin?

2.      Page 8, figure 4: Authors present levels of GSK3b at Figure 4A as well as p-GSK3b. It is clear from pGSK3g image that protein levels are correlating with feshurin addition. Although it is not clear what is the correlation with levels of GSK3b (4th gel from top figure 4a). Why authors present the levels of GSK3b? is there are any significant? since there is not a correlation. More over from the 3 bands represented all species of GSK3b (4th gel from top figure 4a) neither seems to increase (moving from left to right)! Surely one of those should represent the p-GSK3b and thus its levels should change.

Authors.

3.      Page 9, line 217-218: “Following treatment of HaCaT cells with various concentrations (1, 3, and 6 μM) of feshurin for 24, “   Replace  with: of feshurin for 24 h,

4.      Page 11, figure 6B: Figure size should increase it is hardly visible and does not help the reader. There is plenty of space to increase its size.  

5.      Page 11, line 256-262: Authors seem to have docking data for feshurin and PF06700841. They describe the interactions but it will be more useful to present docking score or ΔG non-bonding energies for the two compounds.  

Author Response

Comments1: Page 4, figure 2B: Authors should explain or comment regarding increase of activity in CCK-8 assay after Addition of feshurin in HaCaT cell. How this experiment contribute to Anti-Vitiligo Effects of Feshurin?

Response 1: Thank you for your valuable comments. Epidermal melanocytes and keratinocytes are near each other, forming a functional and structural unit where keratinocytes play a pivotal role in supporting melanocyte homeostasis and melanogenesis. Each melanocyte within the epidermis is in contact with 30–40 neighboring keratinocytes through long dendritic extensions that form adhesive structures. This intimate relationship suggests that keratinocytes might contribute to ongoing melanocyte loss and subsequent depigmentation. IFN-γ induces chemokine production in keratinocytes, promoting initial melanocyte apoptosis, the release of melanocytic-specific antigens, and their presentation to chemokines, thereby attracting and activating CD8+ T cells to initiate adaptive responses. In summary, keratinocytes can influence vitiligo development by a combination of failing to produce survival factors, limiting melanocyte adhesion in lesional skin, presenting melanocyte antigens, and enhancing the recruitment of pathogenic T cells. Keratinocytes control melanocyte proliferation and differentiation not only by soluble mediators but also by direct cell-to-cell contact. One widely utilized human keratinocyte line, HaCaTs, is derived from spontaneously immortalized, human keratinocytes. We selected the safe and effective concentrations of the feshurin by CCK-8 experiments for further studies. No significant cytotoxicity was observed when the cells were treated with 3 μM of IFN-γ and 10 ng/ml of the JAK kinase inhibitor Ruxolitinib alone compared to the concurrent administration of IFN-γ and feshurin.

Comments 2: Page 8, figure 4: Authors present levels of GSK3b at Figure 4A as well as p-GSK3b. It is clear from pGSK3g image that protein levels are correlating with feshurin addition. Although it is not clear what is the correlation with levels of GSK3b (4thgel from top figure 4a). Why authors present the levels of GSK3b? is there are any significant? since there is not a correlation. More over from the 3 bands represented all species of GSK3b (4th gel from top figure 4a) neither seems to increase (moving from left to right)! Surely one of those should represent the p-GSK3b and thus its levels should change.

Response 2: Thanks for your suggestion. Phosphorylated proteins are proteins that have been phosphorylated with a change in protein structure, whereas total proteins are all proteins, including unphosphorylated proteins. By comparing phosphorylated proteins with the corresponding total proteins, changes in protein phosphorylation can be determined more accurately. Feshurin promotes the levels of GSK-3β phosphorylation, phosphorylation is the activated state of the protein.

The melanogenic signal pathways include the Wnt signaling pathway, p38 MAPK, ERK, and JNK MAPK signaling pathway, and the cAMP-dependent signaling pathways. We examined the expression of all signaling pathway proteins associated with melanogenesis. We found that feshurin induced the Akt, GSK-3β, β-catenin-ser 33, 37, 41 phosphorylation, we speculated that the effects of feshurin on melanogenesis might be attributed to the activation of p-Akt that promotes the phosphorylation of GSK-3β, thus increasing the accumulation of β-catenin in the cytoplasm, the accumulated β-catenin is translocated to the nucleus, where it directly binds with MITF to promote its transcription and stimulates the tyrosinase family genes. However, feshurin had no effects on the p38 MAPK, ERK, and JNK MAPK signaling or cAMP-dependent signaling pathways. Our data demonstrated that feshurin induced melanogenesis by activating the Wnt pathway and increasing the expression of the MITF and tyrosinase family genes.

Comments 3: Page 9, line 217-218: “Following treatment of HaCaT cells with various concentrations (1, 3, and 6 μM) of feshurin for 24, “Replace with: of feshurin for 24 h.

Response 3: Thank you for your careful work. We are very sorry for our incorrect writing. we have replaced with:of feshurin for 24 h. (Page 10, line 233)

Comments 4: Page 11, figure 6B: Figure size should increase it is hardly visible and does not help the reader. There is plenty of space to increase its size.

Response 4: Thanks for your suggestion, according to your comments, we have replaced it with a visible large size image. (Page 12, figure 6)

Comments 5: Page 11, line 256-262: Authors seem to have docking data for feshurin and PF06700841. They describe the interactions but it will be more useful to present docking score or ΔG non-bonding energies for the two compounds.

Response 5: Thank you for your valuable advice. The docking score for PF06700841 is

-9.23 kj/mol, and for feshurin is -7.36kj/mol. (Page 12, line 271-272)

Round 2

Reviewer 1 Report

Comments and Suggestions for Authors

Accepted